# Position: Self-Play Only Evolves When Self-Synthetic Pipeline Ensures Learnable Information Gain

**Wei Liu**[1]  **Siya Qi**[1]  **Yali Du**[1 2]  **Yulan He**[1 2]

## Abstract

Large language models (LLMs) make it plausible to build systems that improve through self-evolving loops, but many existing proposals are better understood as self-play and often plateau quickly. A central failure mode is that the loop synthesises more data without increasing *learnable information* for the next iteration. Through experiments on a self-play coding task, we reveal that **sustainable self-evolution requires a self-synthesised data pipeline with learnable information that increases across iterations.** We identify triadic roles that self-evolving LLMs play: the PROPOSER, which generates tasks; the SOLVER, which attempts solutions; and the VERIFIER, which provides training signals, and we identify three system designs that jointly target learnable information gain from this triadic roles perspective. Asymmetric co-evolution closes a weak-to-strong-to-weak loop across roles. Capacity growth expands parameter and inference-time budgets to match rising learnable information. Proactive information seeking introduces external context and new task sources that prevent saturation. Together, these modules provide a measurable, system-level path from brittle self-play dynamics to sustained self-evolution.

## 1. Introduction

The rapid progress of large language models (LLMs) has made self-evolving AI systems plausible (Gao et al., 2025; Fang et al., 2025). In such systems, a model plays different roles to generate data (PROPOSER), produce solutions (SOLVER), and provide feedback signals (VERIFIER), thereby forming self-training loops autonomously. Related

[1]King's College London [2]The Alan Turing Institute. Correspondence to: Wei Liu <wei.4.liu@kcl.ac.uk>, Yulan He <yulan.he@kcl.ac.uk>.

*Proceedings of the 43$^{rd}$ International Conference on Machine Learning*, Seoul, South Korea. PMLR 306, 2026. Copyright 2026 by the author(s).

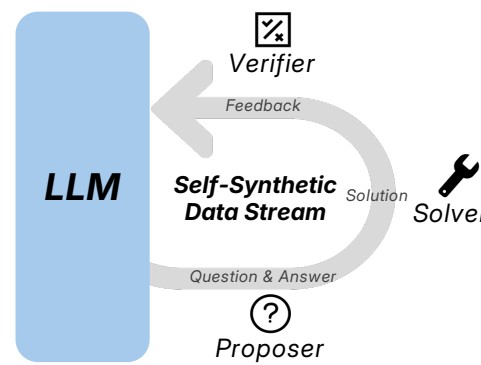

*Figure 1.* A self-evolving LLM plays three roles as PROPOSER, SOLVER and VERIFIER. The whole self-evolving process can be seen as different synthetic operations (synthesis qa, solution and feedback) on the same information source, which is the LLM itself.

research has evolved from early self-supervised training systems (Zelikman et al., 2022; Yuan et al., 2023; Gülçehre et al., 2023; Yuan et al., 2024; Dong et al., 2025; Chen et al., 2024; Qu et al., 2024; Tu et al., 2025) towards more recent self-play systems trained with reinforcement learning (Zhao et al., 2025a; Huang et al., 2025; Yue et al., 2026; Yang et al., 2025; Liu et al., 2025a; Chen et al., 2025a; Lu et al., 2025; Hong et al., 2025a; Wang et al., 2025; Guo et al., 2025b). Some focus on verifiable domains such as mathematics and coding, where a fixed VERIFIER is available, and the LLM performs self-play between the PROPOSER and SOLVER roles. Others emphasise free-form domains, such as preference learning and instruction following, where the SOLVER and VERIFIER are co-evolved on a fixed dataset. In most cases, these systems adopt multi-reward reinforcement learning to achieve self-evolution.

Despite their promise, such systems are often fragile and quickly enter a plateau or collapse after only a few rounds of self-play. In self-play between PROPOSER and SOLVER, Zhao et al. (2025a) report that the PROPOSER tends to generate trivial identity-like problems ($f(x) = x$). Huang et al. (2025) and Yue et al. (2026) observe an early peak and subsequent decline in overall model performance. Chen et al. (2025a) report that the PROPOSER requires carefully tuned prompts to ensure the proposed data remains within a reasonable regime. Some approaches benefit from pe-

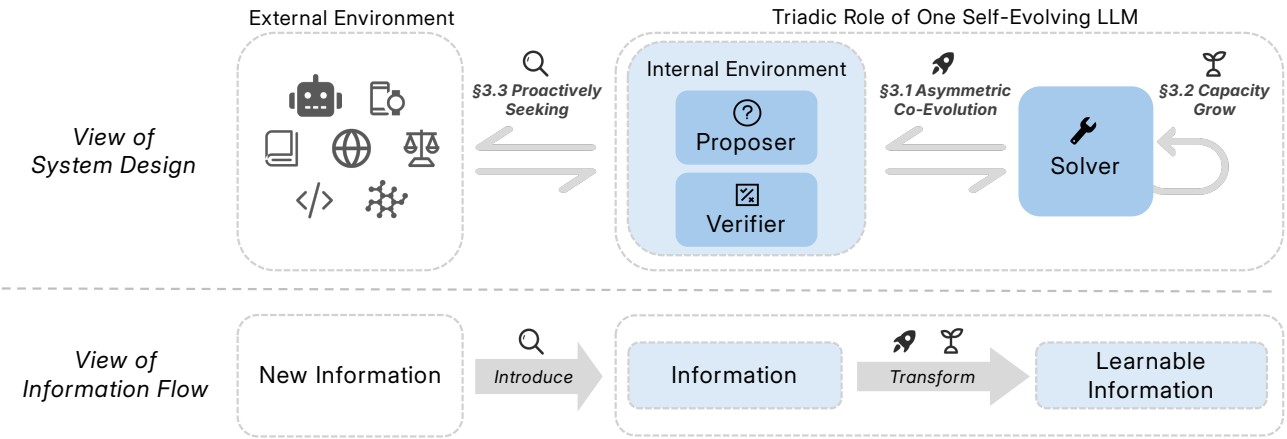

*Figure 2.* Overall framework of a triadic self-evolving loop. A self-evolving LLM plays three roles: the PROPOSER and VERIFIER form the internal environment, proactively interacting with the external environment to provide data and supervision for the SOLVER. The SOLVER and internal environment co-evolve asymmetrically, adaptively expanding capacity to capture more learnable information. From an information perspective, the system continually absorbs external information, and transform them into internal learnable information.

riodically introducing ground truth data to recalibrate the VERIFIER (Yang et al., 2025; Lu et al., 2025). Without such intervention, the system risks falling into a state of self-delusion, resulting in a rapid decline in overall performance.

To resolve this mode collapse and achieve genuine self-evolution, we argue that **self-evolution should be viewed as a healthy self-synthesised data pipeline, where the PRO-POSER, SOLVER, and VERIFIER operate cooperatively to ensure a monotonic increase in learnable information, rather than relying on fragile self-play dynamics.**

This position extends beyond a trivial composition of $P + S$ and $S + V$ into $P + S + V$ (Yang et al., 2025; Chen et al., 2025b). It treats the loop as a data pipeline (as shown in Figure 1) whose success is characterised by increasing *learnable information* during iterations. We make this requirement precise by formalising learnable information under bounded observers in §2.2. The information-theoretic view makes this question concrete. It separates learnable structure from unlearnable noise under bounded computation, which clarifies why reward shaping alone is insufficient for sustained improvement. Empirical analysis of existing self-play loops further validates this diagnosis. Motivated by this observation, we propose three system-level design principles for the self-evolution loop. **1) Asymmetric Co-evolution.** We frame self-evolution as a co-evolutionary process between the SOLVER and its internal environment, consisting of the PROPOSER and VERIFIER. Although all roles share the same underlying weights, their divergent synthetic directions induce an information gap that enables weak-to-strong supervision. This asymmetry needs to be explicitly preserved by synchronising improvements in the SOLVER back to the internal environment as strong-to-weak supervision, thereby sustaining continuous self-improvement. **2) Capac-

ity Growth.** The learnable information in self-synthesised data is determined not only by the data distribution but also by the capacity of the observer model. As self-evolution progresses, the model needs to continually expand its effective *capacity budgets*, encompassing both parameter capacity and inference-time computation, to keep pace with the increasing amount of learnable information produced within the loop. **3) Proactive Information Seeking.** Self-evolution driven by zero-data or a fixed dataset is fundamentally bounded by finite information. We argue that a self-evolving system should proactively acquire external information sources aligned with its current capabilities. Such sources provide not only new contexts for synthetic data but also new synthetic directions, creating fresh asymmetries that can be exploited by the internal co-evolutionary process.

## 2. Background

We model self-evolution as an iterative process that produces self-synthetic training data $D^{(t)}$ and updates a single LLM with it. The goal is not merely to improve task accuracy within a fixed self-play game, but to ensure that the *learnable information* in $D^{(t)}$ increases during iterations.

### 2.1. Background on Triadic Self-Evolution

**View of System Design.** At iteration $t$, the LLM acts as a PROPOSER to generate tasks, optionally conditioned on external context. It then acts as a SOLVER to produce solutions. It finally acts as a VERIFIER to assess the solutions and produce feedback signals. The next update trains the base model on the resulting self-synthetic training instances $D^{(t)}$, which include tasks, solutions, and verification signals. In this loop, there is no external labelled dataset, teacher model,

or reward model, so the loop is self-evolving. The PRO-POSER and VERIFIER form the internal environment (Guo et al., 2025b) of the system. They jointly shape what the SOLVER practises and what feedback becomes a learnable training signal. An external environment, when present, provides information sources such as documents or interactive worlds (Liu et al., 2025a). Usually, such information enters as context that conditions task and evaluation generation. This three-role design matters for scope. $P + S$ with a rule-based VERIFIER constrains its application to reinforcement learning from verifiable rewards (RLVR). $S + V$ focuses on free-form domains and trains a better VERIFIER by learning on fixed preference data instead of self-synthesised data. Triadic self-evolution targets both regimes by evolving all three roles, which share one base model. Improvements in any role update the same underlying model, making the loop self-evolving rather than a collection of separate agents.

**View of Information Flow.** Viewed as information flow, the self-evolving loop produces a self-synthesised data stream. Proposing, solving, and verifying are different transformations of a shared information source: the information contained in the pre-trained weights of the LLM. If proposing is conditioned on an external environment, the loop can incorporate information that is not present in the pre-trained source, which most current self-play systems do not provide. Many self-play systems still achieve progress despite the "no-new-information" concern. This progress can be understood as a transformation that converts unlearnable noise into learnable information for the next update. Based on this view, we introduce three key designs that ensure a three-role self-evolving system supports sustained growth of learnable information across iterations, going beyond fragile reward shaping. We formalise learnable information in §2.2 and then present the three design claims in §3.

### 2.2. Background on Learnable Information

We introduce *learnable information* as the notion that matters for self-evolution. Shannon entropy (Shannon, 1948) characterises the total uncertainty of a distribution. It does not distinguish reusable structure from randomness, which is directly relevant to learning. Minimum description length (MDL) (Rissanen, 1978) provides a criterion that is aligned with learning. It evaluates a model by the description length it induces for the data. This length decomposes into a model description and a prediction loss. This decomposition motivates a separation between learnable structure and unlearnable components. Learnable information refers to the part of data that a learner can capture as a reusable structure, such as patterns that allow better compression or prediction. Unlearnable information is whatever remains unpredictable or incompressible given the learner's assumptions, capacity, and training method, and thus appears as noise. Formal treatments of this distinction are studied extensively in compu-

tational complexity and information theory (Koppel, 1987; Bennett, 1988; McAllister, 2003; Allender et al., 2011).

Importantly, learnable information is not an absolute property of the data, but is defined relative to the observer. While MDL provides a learning-aligned criterion, it assumes an unlimited observer computation budget. However, self-evolving LLMs operate under explicit capacity and computation constraints. Epiplexity (Epistemic Complexity) (Jiang, 2025; Finzi et al., 2026) makes this dependence explicit by introducing computational budgets for the observer. This makes epiplexity a natural match for modelling self-evolving LLMs. We therefore adopt epiplexity as a measurement tool that instantiates an MDL objective under explicit parameter and inference-time budgets, yielding operational proxies for learnable and unlearnable information.

We use $t$ for the self-evolution iteration index. For each role $r \in \{\text{PROPOSER}, \text{SOLVER}, \text{VERIFIER}\}$, we allow budgets $(C_r^{(t)}, T_r^{(t)})$ that can depend on the role and the iteration. Here $C$ denotes parameter capacity and $T$ denotes inference-time computation. When a single budget pair is sufficient, we write $(C, T)$. We denote by $D_d^{(t)}$ the distribution over synthetic directions $d$ used by the internal environment to generate self-synthetic training instances at iteration $t$. Let $X$ denote the self-synthesised data stream produced by the loop. Following Finzi et al. (2026), we make the bounded observer constraints explicit: let $C$ denote a parameter budget (capacity) and let $T$ denote an inference-time budget (computation/trajectory length). Let $\mathcal{P}_{C,T}$ denote a family of LLM observers implementable within budgets $(C, T)$. Define the bounded MDL optimiser:

$$\text{P}^\star = \underset{\text{P} \in \mathcal{P}_{C,T}}{\arg\min} \left\{ |\text{P}| + \mathbb{E}\left[\log \frac{1}{P(X)}\right] \right\}, \quad (1)$$

and define $\text{S}_{C,T}(X)$ and $\text{H}_{C,T}(X)$ as:

$$\text{S}_{C,T}(X) := |\text{P}^\star|, \quad (2)$$

$$\text{H}_{C,T}(X) := \mathbb{E}\left[\log \frac{1}{P^\star(X)}\right], \quad (3)$$

$$\text{MDL}_{C,T}(X) := \text{S}_{C,T}(X) + \text{H}_{C,T}(X). \quad (4)$$

where $\text{S}_{C,T}(X)$ is the *epiplexity*, and $\text{H}_{C,T}(X)$ is the *bounded entropy*. Intuitively, $\text{H}_{C,T}(X)$ represents what still appears random to the bounded observer, which can be regarded as unlearnable information given an LLM constrained by $(C, T)$. In contrast, $\text{S}_{C,T}(X)$ represents the reusable structure the observer needs to internalise to compress or predict the data, which we treat as a proxy for learnable information. Because these quantities depend on $(C, T)$, the same object can appear structured to a stronger observer and random to a weaker one. Epiplexity thus identifies a "Goldilocks Zone" for self-evolution: data must be neither too simple (low S, low H) nor too hard (low S, high

H) for the current observer. Sustainable progress requires the system to continuously generate data within this zone, where *the structure is complex enough to be non-trivial but structured enough to be learnable*.

## 3. Towards Genuine Self-Evolution

We now connect the self-evolving loop to three necessary designs: **asymmetry**, **capacity**, and **information seeking**, as shown in Figure 2. For each claim, we separate the system-level mechanism from its information-theoretic justification. We then contrast these mechanisms with existing self-play systems and summarise practical implementations.

### 3.1. Asymmetric Co-evolution

**Design.** In many tasks, proposing and verifying are substantially easier than solving. This asymmetry is well established in easy-to-verify domains such as mathematics and coding (Guo et al., 2025a), and also appears in more general task settings (Burns et al., 2024). Leveraging this asymmetry enables a self-evolving system. When a single LLM plays all three roles, its current proposing and verification ability can supervise the training of a stronger SOLVER (weak-to-strong). For sustained self-evolution, the improved SOLVER needs to be synchronised back into the internal environment (strong-to-weak) to close the loop, so that proposing and verification keep pace with the SOLVER frontier, as illustrated in Figure 4. In contrast, using a stronger LLM to propose and verify to train a weaker LLM constitutes distillation rather than self-evolution. Reinforcement learning can realise the weak-to-strong transition. We argue for a more explicit design of the strong-to-weak directions, with two objectives: ensure that capacity of **VERIFIER** scale with the solver; ensure that **PROPOSER** remains at the SOLVER frontier and continues to open new synthetic directions. This suggests that asymmetry should be organised as a progressive ladder matched to the solver's frontier.

**Information Perspective.** All three roles operate on the same information source, namely the pre-trained weights, but along different synthetic directions. From a fixed source, such transformations do not create new Shannon information (Shannon, 1948), but they can redistribute learnable information across forward and inverse directions under bounded computation when the inverse mapping is computationally hard (Finzi et al., 2026). In our setting, a synthetic direction $d(P, S, V)$ defines a computation that maps shared weights into a data stream $X_d$. Consequently, the bounded MDL decomposition into $S_{C,T}(X_d)$ and $H_{C,T}(X_d)$ can differ across roles, even under identical resource budgets.

Beyond information creation, boundedness also breaks the *symmetry of information* across factorisations, inducing a gap between forward and inverse directions that naturally

aligns with proposing and verifying versus solving. Although not all tasks can be strictly modelled as a function $Y = f(X)$ and its inverse, the underlying computational asymmetry is broadly applicable. The PROPOSER maps pre-trained knowledge into a problem space, such as generating a poem topic or a mathematics problem, while the SOLVER maps that problem into a concrete solution trace. Even in open-ended tasks like creative writing, generating a high-level constraint ("write a poem about spring") is computationally cheaper than producing a specific instance that satisfies it. A one-way permutation illustrates how extreme this asymmetry can be. Given $f$ as a polynomial-time computable one-way permutation secure against non-uniform Probabilistic Polynomial-Time (PPT) inverters with negligible success probability, if we apply it to uniform input $X$ to produce $Y = f(X)$, then

$$H_{\text{poly}}(X|Y) - H_{\text{poly}}(Y|X) \geq c \log n \qquad (5)$$

The gap of $\Omega(\log n)$ bits quantifies how much harder it is to predict backwards versus forward (Finzi et al., 2026). This gap captures the information-theoretic form of intelligence demand asymmetry. The internal environment can cheaply generate and verify instances, while the SOLVER needs to expend additional computation to reduce residual uncertainty in the inverse direction. This perspective also clarifies the role of gap size. A larger directional gap can expose richer structure and increase potential epiplexity gain, but when the gap exceeds the SOLVER capacity, it appears as time-bounded randomness, yielding noise-like tasks and stalled learning. Therefore, self-evolution requires an *asymmetry ladder* that matches the gap to the current SOLVER, together with strong-to-weak synchronisation so the PROPOSER and VERIFIER track the SOLVER frontier.

**Gaps.** Standard reinforcement learning can achieve the weak-to-strong flow from the PROPOSER/VERIFIER to SOLVER, but it is less clear whether improvements in the SOLVER reliably induce corresponding gains in the PROPOSER and VERIFIER. If the internal environment fails to co-evolve, then as the SOLVER improves, the task/feedback stream can become low-structure relative to the current observer, encouraging collapse towards trivial data.

**Practice.** To fully exploit asymmetry in self-evolving systems and to close the loop of weak-to-strong and strong-to-weak stably, it requires both data engineering and algorithmic innovations: **1) Organise self-synthetic directions by asymmetry gaps.** self-synthesised data need to be carefully structured, allowing the LLM to grow by climbing an asymmetry ladder, progressing from small gaps to large gaps, and eventually to reverse gaps (when the PROPOSER and VERIFIER become sufficiently strong). This requires organising synthetic directions by domain (e.g., mathematics problems typically exhibit a large gap; grammar correction has minimum gaps; and healthcare tasks may exhibit

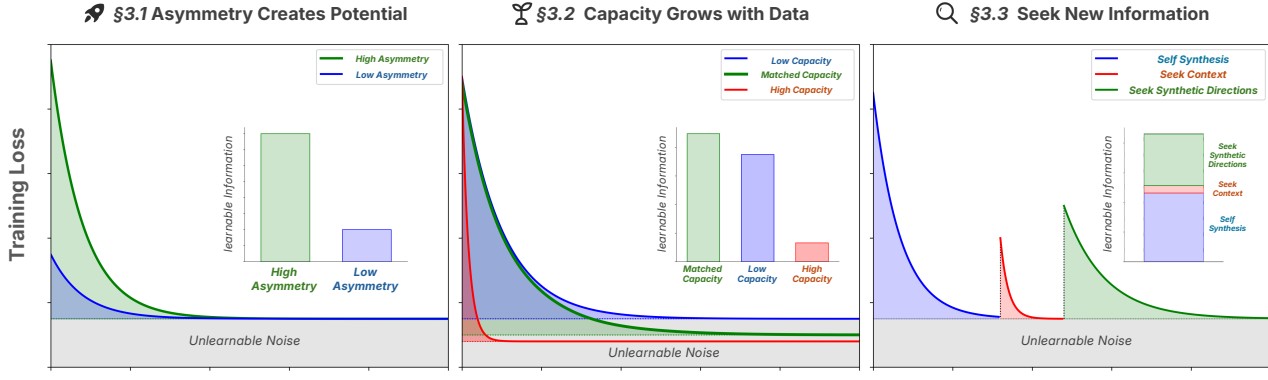

*Figure 3.* Illustration of three designs from the perspective of learnable information. Asymmetry between the SOLVER and the PROPOSER/VERIFIER creates learning opportunities. Expanding model capacity to match self-evolving data opens space for learnable information. Reusing the same patterns in new contexts yields limited gains, whereas introducing new synthetic directions creates fresh asymmetries and thus new sources of learnable information.

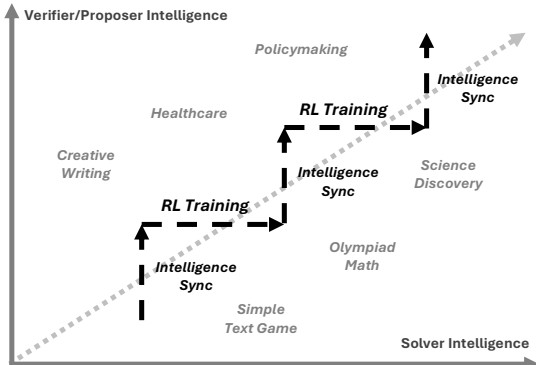

*Figure 4.* Climbing the intelligence asymmetry ladder by closing the loop among PROPOSER, SOLVER, and VERIFIER. "Intelligence synchronisation" denotes updating the weaker PROPOSER/VERIFIER with strong SOLVER. "Reinforcement learning" uses the weaker PROPOSER/VERIFIER to train the SOLVER.

inverse gaps), and even within domains (e.g., abduction, induction, and deduction problems in coding (Zhao et al., 2025a), or conjecture proving vs. large-number multiplication vs. Sudoku in maths). **2) Implement strong-to-weak synchronisation for the PROPOSER.** For the PROPOSER, self-play reward designs, such as using a 50% pass rate for the SOLVER as a reward, can partially address synchronisation, though multi-reward training is often unstable (Zhao et al., 2025a; Kwan et al., 2025; Huang et al., 2025; Chen et al., 2025b). An alternative approach is to back-translate higher-quality PROPOSER's data from a stronger SOLVER. For example, Magicoder (Wei et al., 2024) generates diverse instructions from code snippets; MathGenie (Lu et al., 2024) back-translates questions from augmented ground truth; InverseCoder (Wu et al., 2025b) summarises code into instructions. **3) Implement strong-to-weak synchronisation for**

**the VERIFIER.** For the VERIFIER, reward design is more challenging; most research uses self-consistency or internal belief signals (Zhao et al., 2025b; Zuo et al., 2025; Yang et al., 2025), which do not guarantee improvement of the verifier. A promising direction is verifier-free RL (Liu et al., 2025b; Yu et al., 2025; Zhou et al., 2025), where the VERIFIER and SOLVER explicitly share the same optimisation objective, typically maximising the probability of generating the ground truth answer given the question and model's reasoning trajectory.

### 3.2. Capacity Budgets Grow Across Iterations

**Design.** Most self-play loops fix the observer across iterations. Sustainable self-evolution instead requires budgets that grow with iteration. We define *capacity* as the portion of the model effectively participating in learnable information extraction, which may include the full parameter set, or a sparsely activated component (e.g., experts, layers). In general, capacity is determined by a parameter budget $C^{(t)}$, an inference-time budget $T^{(t)}$, and a training budget $B^{(t)}$. In many existing self-evolving setups, the incremental dataset per iteration is small, so $B^{(t)}$ remains almost constant. However, $C^{(t)}$ and $T^{(t)}$ should not be fixed: as the loop exposes more learnable structure over time, capacity should increase to allow the observer to absorb new information, either by growing parameters, activating more of the model, or increasing reasoning length.

**Information Perspective.** Epiplexity makes the dependence on the observer budgets $(C, T)$ explicit by defining $S_{C,T}(X)$ and $H_{C,T}(X)$ through an MDL objective optimised within an observer family $\mathcal{P}_{C,T}$. Fixing $(C, T)$ bounds the amount of reusable structure the observer can internalise from the self-synthetic stream. Expanding these budgets enlarges the observer family. If $\mathcal{P}_{C_1,T_1} \subseteq \mathcal{P}_{C_2,T_2}$,

then $\mathrm{MDL}_{C_2,T_2}(X) \leq \mathrm{MDL}_{C_1,T_1}(X)$. Budget expansion, therefore, shifts the boundary between reusable structure and residual randomness.

**Gaps.** Budget mismatches induce characteristic failure modes. Fixing $C^{(t)}$ while the loop generates richer trajectories prevents the observer from representing the abstractions needed to compress new data. Training loss saturates and progress plateaus. The loop then shifts toward directions that remain easy for the current model class, reducing frontier pressure and potentially collapsing to trivial tasks (Huang et al., 2025; Yue et al., 2026). Fixing $T^{(t)}$ produces a distinct failure mode. The internal environment can raise difficulty by proposing tasks that require longer reasoning chains or more tool use. When the inference-time budget is limited, errors stem from truncated inference rather than learnable deficiencies. Both mismatches reduce the learnable information available to subsequent updates.

**Practice.** Capacity growth should be planned along both axes. The parameter budget $C^{(t)}$ can be expanded through role-asymmetric scaling where a smaller PROPOSER/VERIFIER trains a larger SOLVER, and the internal environment is refreshed from the stronger checkpoint. Alternatively, the parameter budget can be expanded by adding parameters across iterations, allowing the base model itself to grow without relying on a larger pre-trained model (Gong et al., 2019; Xie et al., 2020; Hong et al., 2025b; Singh et al., 2025). It is also worth exploring activated subset growing (Huang et al., 2024; Nishu et al., 2025). The inference-time budget $T^{(t)}$ should likewise be explicit and dynamic. The system should allocate increasing computation per instance as iteration advances. This requires adaptive reasoning along the inference token axis (Alomrani et al., 2025; Qu et al., 2025) or adaptive recursive depth along the layer axis (Bae et al., 2025).

### 3.3. Proactive Information Seeking

**Design.** A closed self-play loop without external interaction is bounded by information already present in the current system. Simply adding a fixed external corpus does not resolve this limitation because the loop collapses into repeated training on a static support. We therefore treat information seeking as an explicit responsibility of the internal environment (PROPOSER+VERIFIER): at each iteration, it should select external contexts, and learn new synthetic directions around them.

**Information Perspective.** External information can enter a self-evolving loop in two ways. It can be incorporated directly into training or supervision, which increases total information but breaks pure self-synthesis. Alternatively, it can be used only as a conditioning context. Let $d^{(t)}$ denote the external context obtained at iteration $t$, and $Y^{(t)}$ as self-synthetic outputs generated by the internal environment

conditioned on $d^{(t)}$. Since optimisation targets lie in $Y^{(t)}$ rather than $d^{(t)}$, the relevant quantity is the learnable information in the conditional stream $(Y^{(t)} \mid d^{(t)})$. We formalise this using a conditional bounded MDL objective (for brevity, we denote $MDL_{C,T}$ as $MDL$):

$$\mathrm{MDL}(Y \mid d) := \min_{P \in \mathcal{P}_{C,T}} \left\{ |P| + \mathbb{E}\left[\log \frac{1}{P(Y \mid d)}\right] \right\} \quad (6)$$

, and treat $\mathrm{S}_{C,T}(Y \mid d)$ as conditional learnable information by analogy with Equation (2). Even with static $d^{(t)}$, the internal environment can repeatedly propose new questions and evaluations that demand progressively richer reasoning. This shifts the synthetic direction support without using the corpus as labels. The mechanism is theoretically supported by conditional epiplexity, which preserves information gain and asymmetry properties, and is empirically supported by self-play in corpus environments (Liu et al., 2025a). Finally, since epiplexity depends on factorisation and ordering (Finzi et al., 2026), how context is converted into tasks, and the schedule by which tasks are introduced, directly affect which structure becomes learnable at each iteration.

**Gaps.** Existing systems typically fall into three regimes. Zero-data systems use no external information, so reachable contexts are confined to the current weights and the loop recycles directions. Dataset-driven systems sample from a fixed corpus (Liu et al., 2025a), which reduces to fine-tuning on the corpus, and limits the internal environment from catching the evolving SOLVER frontier. A third class attaches external context via a fixed, iteration-independent mechanism (Xu et al., 2025; Zhang et al., 2025a), yielding a static context distribution. Early in training, such contexts often exceed the SOLVER budgets and yield little learning. Later, the same mechanisms become routine and fail to expose new learnable structure. Across regimes, information seeking is reactive rather than proactive: systems consume available context instead of selecting contexts and transformations aligned with the current frontier.

**Practice.** Proactive information seeking can be realised as an adaptive policy within the internal environment: **1) Learn to ask for information.** The PROPOSER generates queries from SOLVER failures, VERIFIER disagreement, or persistent error patterns, and after retrieving $d$, synthesises tasks whose solutions require explicit use of $d$, such as citation grounded answers, multi-document synthesis, or contradiction detection, thereby coupling retrieval to the current frontier rather than treating it as fixed preprocessing. **2) Turn context into asymmetry gaps rather than hints.** Given the same retrieved $d$, the internal environment synthesises multiple synthetic directions with different difficulty profiles, and schedules them as a curriculum matched to the iteration state, prioritising grounding and evaluation early and inverse or compositional directions later as budgets grow, which exploits factorisation

dependence of bounded information and sustains learnable information growth. **3) Co-evolve with the external environment.** Retrieval, reranking, and memory are treated as evolving components and updated using self-synthetic signals such as verifier-based relevance checks; whereas prior work co-evolves only the SOLVER with the internal environment (Guo et al., 2025b) or frames external evolution as memory refinement over trajectories (Hu et al., 2025).

### 3.4. Synergy

These three modules are not independent add-ons but functional components of a single *information production pipeline* whose synergy ensures a monotonic increase in learnable information. Specifically, **Asymmetric Co-evolution** acts as the *generator* by exploiting the computational gap between verifying and solving, it performs the critical transformation of converting unlearnable noise into learnable structure, creating the "information potential" that drives the solver's gradient. To capture this gain, **Capacity Growth** acts as the *receiver*, expanding the observer's hypothesis space to internalise newly exposed learnable information that would otherwise saturate a fixed-budget model. Finally, **Proactive Information Seeking** acts as the *open feeder*, continuously injecting fresh entropy and contexts into the internal environment to ensure the generator never exhausts its raw material. Together, these designs transform self-evolution from a finite game of self-play into an open-ended process of learnable information discovery, as shown in Figure 3.

## 4. Experiments

We conduct small-scale self-play training experiments and extend the Prequential Coding-based estimation method of Finzi et al. (2026) to estimate learnable information (see algorithm details in Appendix B). The experiments are diagnostic rather than exhaustive, aiming to illustrate how learnable information varies across roles, capacities, and synthetic directions. We use the prequential code length at the point achieving the optimal MDL value to estimate epiplexity as a measure of learnable information, which is

$$|\mathrm{P}_{\mathrm{preq}}| \approx \sum_{i=0}^{M-1} \left( \log \frac{1}{P_i(Z_i)} - \log \frac{1}{P_M(Z_i)} \right), \quad (7)$$

Here, $Z_i$ is the $i$-th training token, $P_i(\cdot)$ denotes the predictive distribution of the model before observing $Z_i$, and $P_M(\cdot)$ is the predictive distribution of the fully trained model. This quantity can be interpreted as the effort expended by the model to learn the data. Further details are provided in Appendix A. We conduct two experiments. **Experiment 1** compares the epiplexity values observed by solvers of different capacities on synthetic data generated by proposers of varying capacities along different synthetic

data directions. **Experiment 2** examines how the synthetic data evolves over iterations as the model engages in continued self-play. The data and self-play setup follows Zhao et al. (2025a), which has three types of code-based tasks, including abduction, where the input is generated given a program and its output; deduction, where the output is generated given a program and an input; and induction, where the program is generated given an input and an output. Examples of the three task types are shown in Appendix E.

**Experiment 1.** The results are shown in Figure 5, from which we observe the following. 1) Stronger proposers, from Qwen2.5 7B to Qwen2.5 14B to Qwen3 4B, generate synthetic data that contains a larger amount of learnable information. 2) As the SOLVER size increases, the learnable information first increases and then decreases. This is consistent with the emergence phenomena observed in Finzi et al. (2026). Under a fixed computation budget, the model is forced to learn compressible patterns and structure, in which regime a larger model can observe higher learnable information. Once a certain budget threshold is exceeded, the model instead opts for direct memorisation and abandons learning effective structure, leading to a decrease in learnable information. 3) Different synthesis directions yield different amounts of learnable information, with induction being substantially higher than abduction and deduction, which aligns with intuition. These preliminary experiments support our claim: different synthetic directions yield varying amounts of information, and effective co-evolution is necessary to continuously increase learnable information; otherwise, simply increasing the proposer's capacity may in fact reduce the information content.

**Experiment 2.** The results are shown in Figure 6. After multiple iterations of self-play training, we observe that the amount of information does not increase steadily but instead fluctuates dramatically. This aligns with both the Zhao et al. (2025a) and our empirical observations. Without an explicit mechanism to close the self-training loop and relying solely on multi-reward reinforcement learning, the model fails to achieve sustained evolution. Behaviourally, this manifests as a decline in SOLVER capability and a collapse of the problem patterns generated by the proposer.

## 5. Alternative Views

Several perspectives have been proposed to explain progress in self-evolving language model systems. However, we argue that none of these perspectives provides a sufficient criterion for identifying genuine and sustained self-evolution.

**Self-Evolving via Self-Play RL.** Self-evolution is often studied with stable reward design and monotonic improvement under reinforcement learning. While reward optimisation is essential for guiding behaviour and stabilising train-

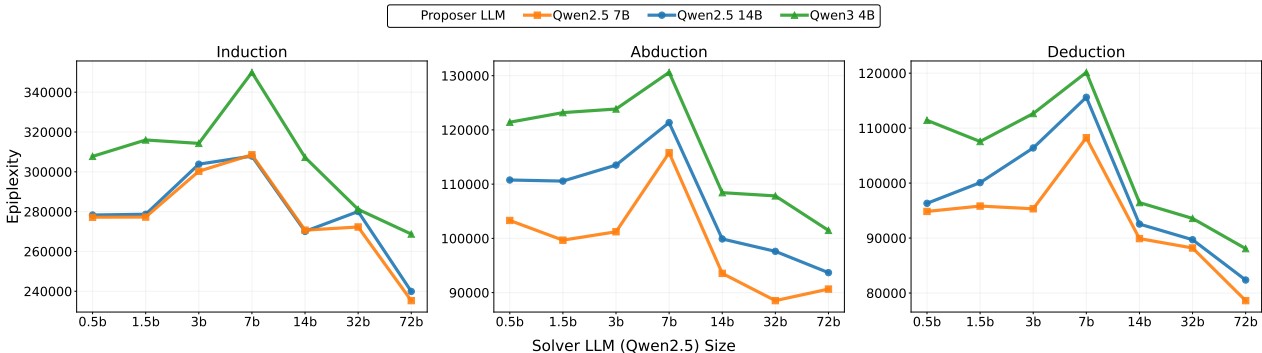

*Figure 5.* Epiplexity results on synthetic data with different tasks (induction, abduction and deduction) proposed by different PROPOSER LLMs and observed by different SOLVER LLMs. See details of calculating epiplexity in Appendix C.

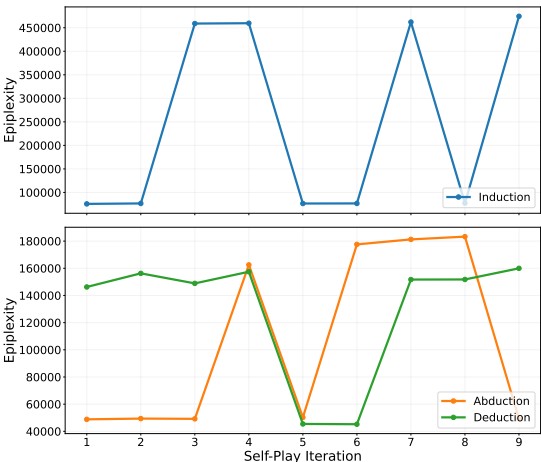

*Figure 6.* Epiplexity results during the self-play training on three tasks. See task detail in Appendix E.

ing, it does not ensure that the self-synthesised data stream exposes an increasing learnable structure under bounded observers. Systems may improve reward via hacking (Jiang et al., 2025; Zha et al., 2025; Zhang et al., 2025b), rely on memorised pre training knowledge rather than reasoning (Shao et al., 2025; Wu et al., 2025a; Yan et al., 2026), or exhibit instability in multi-reward self-play (Zhao et al., 2025a; Kwan et al., 2025; Huang et al., 2025; Chen et al., 2025b). In these cases, task-level metrics improve while learnable information remains unchanged. We therefore view reward optimisation as necessary but not sufficient for genuine self-evolution.

**Curriculum Learning for Evolution.** Curriculum-based approaches aim to match task difficulty to the solver's capability to sustain learning and avoid collapse. While essential, difficulty conflates factors such as search depth and verification cost and does not directly reflect whether a new structure becomes learnable. A curriculum can increase apparent difficulty while repeatedly sampling structurally

similar synthetic directions, leading to optimisation without information gain. From an information perspective, curricula contribute to self-evolution only when they increase bounded learnable information.

**Co-Evolution between Agent and Environment** Co-evolutionary perspectives frame progress as mutual adaptation between an agent and an evolving environment (Guo et al., 2025b; Wang et al., 2019). Our framework is compatible with this view and models the proposer and verifier as an internal environment that co-evolves with the solver. However, co-evolution alone does not distinguish productive adaptation from dynamics that reshuffle complexity without expanding internal structure. An evolving environment can become more challenging while exposing less reusable structure to a bounded learner. Introducing an explicit notion of learnable information provides a principled way to assess when co-evolution supports sustained self-evolution.

**Scaling is All you Need.** Increasing model size or inference-time budget is often expected to resolve stagnation in self-evolving systems. Greater capacity expands what can be learned. However, without mechanisms that introduce new synthetic directions and maintain asymmetry, increased capacity may primarily amplify memorisation rather than structural generalisation. We therefore view capacity scaling as a necessary component coordinated with information growth, rather than a sufficient explanation on its own.

## 6. Related Works

**Self-Training.** Early self-training methods bootstrap reasoning from self-generated data with fixed verification: STaR (Zelikman et al., 2022) filters correct reasoning traces and recycles rationalised ones for fine-tuning, ReST (Gülçehre et al., 2023) alternates offline generation and reward filtered updates, and scaling studies (Yuan et al., 2023) report log-linear gains from rejection sampling. These approaches saturate once the initial distribution is exhausted.

**Solver–Verifier Co-Evolution.** Self-rewarding (Yuan et al., 2024) uses the model as its own judge via LLM-as-a-judge prompting; SPIN (Chen et al., 2024) distinguishes synthetic from human responses through iterative DPO; Self-boosting (Dong et al., 2025) generates diverse prompts and improves responses iteratively; iterative DPO (Tu et al., 2025) refines both generator and reward model over multiple rounds with verifiable rewards; URPO (Lu et al., 2025) unifies policy and reward optimisation in a unified task format; Cooper (Hong et al., 2025a) co-optimises both using hybrid rule-based and model-based rewards to prevent reward hacking; Recursive introspection (Qu et al., 2024) frames multi-turn self-correction as an MDP. However, these methods lack explicit strong-to-weak synchronisation to ensure the VERIFIER tracks SOLVER improvements or expands the task distribution beyond the initial corpus.

**Proposer–Solver Self-Play.** Absolute Zero (Zhao et al., 2025a) self-propose code-based tasks verified by execution; R-Zero (Huang et al., 2025) uses pseudo-labels from majority voting and rewards tasks near the solver's decision boundary; Dr. Zero (Yue et al., 2026) applies hop-grouped policy optimisation to search agents with multi-turn reasoning; Self-Questioning (Chen et al., 2025a) generates topic-conditioned questions verified via majority voting or unit tests. These methods achieve rapid initial gains but report instability and collapse after a few iterations because the PROPOSER drifts towards trivial or unsolvable tasks.

**Triadic Loops** SPELL (Yang et al., 2025) introduces a questioner–responder–verifier loop for long-context reasoning with curriculum over context length; SPICE (Liu et al., 2025a) grounds task generation in external corpora to provide information asymmetry and reduce hallucination; Socratic-Zero (Wang et al., 2025) uses a frozen teacher to verify and craft novel questions targeting SOLVER weaknesses; GenEnv (Guo et al., 2025b) co-evolves the policy and internal environment that generates tasks aligned to the agent's current ability. Despite these advances, most systems still report early plateaus, require careful reward tuning, or lack a unified principle for diagnosing why loops stall. Our position addresses these gaps by framing self-evolution as a learnable information pipeline under bounded observers, making three system-level requirements explicit.

## 7. Limitations

This work provides a preliminary approach to designing a self-evolving framework for stable growth of learnable information, yet it remains far from a fully mature, off-the-shelf solution, and several limitations warrant further exploration. First, closing the asymmetry gap is currently applicable only in domains that are easy to verify. On the hard-to-verify side, more cross-domain, generalizable methods are needed to make breakthroughs. Second, learnable information can-

not replace metrics related to final task accuracy, as it is a macroscopic measure; not all learnable information is necessarily useful for task completion, as it may primarily reflect the structure inherent in the data. Both types of metrics need to be considered together to achieve comprehensive monitoring of self-evolving systems. What's more, the epiplexity metric comes from recent work (Finzi et al., 2026) and isn't yet widely validated. Finally, realising proactive context information seeking remains a major challenge, as it requires the model to recognise what it does not know (Yin et al., 2023) and explicitly formulate it as a query, which is an inherently difficult research problem.

## 8. Call to Action

We urge the research community to shift focus from optimising static self-play loops to designing dynamic self-synthetic pipelines that guarantee monotonic *learnable information gain*. To achieve sustainable self-evolution, future systems must integrate three essential mechanisms: (1) **Asymmetric Co-evolution** to continuously exploit the computational gap between verifying and solving; (2) **Capacity Growth** to expand parameter and inference budgets commensurate with rising structural complexity; and (3) **Proactive Information Seeking** to inject fresh context into the internal environment. We propose evaluating progress not solely by downstream accuracy, but by the system's capacity to discover and internalise new structure, quantified by bounded observer metrics such as epiplexity. In Section 3, we provide detailed *Practice* paragraphs for each component to describe currently feasible directions. However, realising a truly robust and usable self-evolving system still requires overcoming numerous challenges in the coordinated design and implementation of models, data, algorithms, and infrastructure. We call on the community to join efforts toward building genuinely sustainable self-evolving systems.

## 9. Conclusion

Prevailing stagnation in self-evolving systems stems not from insufficient reward optimisation, but from the failure to sustain a monotonic increase in learnable information for bounded observers. By reframing self-evolution as a dynamic self-synthetic data pipeline rather than a static reinforcement learning game, we clarify that sustainable progress requires a loop with asymmetric co-evolution, dynamic capacity expansion and proactively information seeking. Ultimately, this information-theoretic perspective provides the necessary system-level principles to transform fragile self-play dynamics into robust, continuous self-evolution.

## Acknowledgments

This work was supported in part by the UK Engineering and Physical Sciences Research Council through a Turing AI Fellowship (grant no. EP/V020579/1, EP/V020579/2) and the Prosperity Partnership scheme (grant no. UKRI566). Wei is supported by a PhD studentship provided by King's College London (KCL). The authors acknowledge the use of Computational Research, Engineering and Technology Environment (CREATE) at KCL.

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

## A. Experiment Setup

We adopt the publicly available Absolute Zero repository and perform LoRA fine-tuning (rank=16, alpha=32, dropout=0.05) on the Qwen2.5 family of models, spanning from 0.5B to 14B parameters. The training is conducted on seed datasets generated by Qwen2.5 7B, Qwen2.5 14B, and Qwen3 4B. For evaluation purposes, 10% of each dataset is randomly set aside as a validation set, which is used to calculate the bounded entropy as defined in Equation 3.

All experiments use a maximum sequence length of 2048 tokens, with a learning rate of 1e-4. Training is limited to 20 epochs, and both MDL and prequential code lengths are computed at the end of each epoch. The prequential code length corresponding to the minimum MDL is then taken as the model's epiplexity. Early stopping is employed if the MDL fails to decrease for five consecutive epochs; in practice, all runs terminate before reaching the 20-epoch limit.

For the self-play experiments, we follow the Absolute Zero reinforcement learning algorithm, using the same hyperparameters as in the public repository, and train a Qwen2.5 3B model. This model is subsequently used for epiplexity calculations as well.

## B. Epiplexity Estimation Algorithm

Algorithm 1 presents the pseudocode for estimating epiplexity using prequential coding.

The algorithm estimates the epiplexity of a pre-trained model using a prequential Minimum Description Length approach. The dataset $\mathcal{D}$ is split into training and validation subsets. During training, the model processes each batch sequentially, updating parameters via gradient descent. In the first pass, the prequential loss is accumulated to measure the difficulty of learning each data point. At the end of each epoch, the total training and validation losses are computed. The epiplexity is defined as the difference between the prequential and final training losses, representing the model's cumulative online regret. The MDL score combines the normalised epiplexity (model cost) and validation loss per-token (data cost). The algorithm then returns the epiplexity corresponding to the epoch that minimises this MDL score.

---

**Algorithm 1** Epiplexity Estimation via Prequential MDL

---

**Require:** Observer LLM $M_{\theta_0}$, dataset $\mathcal{D} = \mathcal{D}_{\text{train}} \cup \mathcal{D}_{\text{val}}$
**Require:** Max epochs $K$, learning rate $\eta$
**Ensure:** Observed epiplexity $\mathcal{E}^*$ from $M_{\theta_0}$ on $\mathcal{D}$
1: $\theta \leftarrow \theta_0$
2: $\mathcal{L}_{\text{online}} \leftarrow 0$, $N_{\text{train}} \leftarrow 0$
3: $MDL^* \leftarrow \infty$, $\mathcal{E}^* \leftarrow 0$
4: **for** $k = 1$ **to** $K$ **do**
5:     **for** $x \in \mathcal{D}_{\text{train}}$ **do**
6:         $\ell \leftarrow -\log P_\theta(x)$
7:         **if** $k = 1$ **then**
8:             $\mathcal{L}_{\text{online}} \leftarrow \mathcal{L}_{\text{online}} + \ell$
9:             $N_{\text{train}} \leftarrow N_{\text{train}} + \text{CountTokens}(x)$
10:         **end if**
11:         $\theta \leftarrow \theta - \eta\nabla_\theta \ell$
12:     **end for**
13:     $\mathcal{L}_{\text{train}} \leftarrow \sum_{x \in \mathcal{D}_{\text{train}}} -\log P_\theta(x)$
14:     $\mathcal{L}_{\text{val}} \leftarrow \sum_{x \in \mathcal{D}_{\text{val}}} -\log P_\theta(x)$
15:     $N_{\text{val}} \leftarrow \text{CountTokens}(\mathcal{D}_{\text{val}})$
16:     $S \leftarrow (\mathcal{L}_{\text{online}} - \mathcal{L}_{\text{train}})/\ln 2$
17:     $MDL \leftarrow S/N_{\text{train}} + (\mathcal{L}_{\text{val}}/\ln 2)/N_{\text{val}}$
18:     **if** $MDL < MDL^*$ **then**
19:         $MDL^* \leftarrow MDL$
20:         $\mathcal{E}^* \leftarrow S$
21:     **end if**
22: **end for**
    **return** $\mathcal{E}^*$

---

## C. Epiplexity Calculation Details

Figures A1, A2, and A3 respectively illustrate the computation of epiplexity shown in Figure 5. Given a synthetic dataset generated by the PROPOSER, we train a SOLVER model on this data and use the prequential coded length at the checkpoint that achieves the optimal MDL as the epiplexity. As shown, although the entropy decreases monotonically and the coded length increases monotonically, all experimental runs attain the optimal MDL at an intermediate stage of training.

Due to the relatively small data scale in our proof of concept experiments, we adopt an improved computation scheme compared with Finzi et al. (2026). All tokens within the same batch are assigned an identical training loss value, and per-token averages are used when computing the prequential coded length $|P_{\mathrm{preq}}|$ and entropy $H$. These averaged quantities are then summed to obtain a per-token $MDL$ value, which is used to determine the $MDL$ optimal point.

In addition, the epiplexity computation in Finzi et al. (2026) assumes a sufficiently large dataset, such that training for a single epoch suffices and the final loss $-\log P_M(Z_{last})$ can be used as an approximation of the loss $-\log P_M(Z_i)$ incurred by the trained observer on each data point. In our setting, the dataset is relatively small, so we adopt a more accurate procedure by re-computing the loss for each training example after the observer has been fully trained. Furthermore, for multi-epoch training, we define the prequential coded length as the difference between the training loss in the first epoch and the loss of the model trained for multiple epochs, evaluated on all data points, to satisfy the assumptions underlying prequential coding.

## D. Reward and Epiplexity Across Iterations

To clarify the relationship between reward and Epiplexity, we report the proposer reward, solver reward, and Epiplexity together across the nine iterations of Experiment 2. The results are presented in Table A1.

*Table A1.* Proposer reward, solver reward, and Epiplexity ($\times 10^3$ bits) across nine iterations of Experiment 2, reported separately for Induction, Abduction, and Deduction tasks.

| | Induction | | | Abduction | | | Deduction | | |
|---|---|---|---|---|---|---|---|---|---|
| Iteration | Proposer Reward | Solver Reward | Epiplexity | Proposer Reward | Solver Reward | Epiplexity | Proposer Reward | Solver Reward | Epiplexity |
| 1 | −0.781 | −0.296 | 75.7 | −0.152 | 0.147 | 48.8 | −0.065 | −0.036 | 146.2 |
| 2 | −0.478 | −0.282 | 76.7 | −0.181 | 0.569 | 49.4 | 0.041 | 0.273 | 156.3 |
| 3 | −0.428 | −0.273 | 458.9 | −0.297 | 0.733 | 49.2 | 0.045 | 0.240 | 148.9 |
| 4 | −0.368 | −0.227 | 459.6 | −0.320 | 0.775 | 162.5 | 0.124 | 0.217 | 157.4 |
| 5 | −0.332 | −0.157 | 76.5 | −0.356 | 0.847 | 50.2 | 0.091 | 0.266 | 45.4 |
| 6 | −0.360 | −0.197 | 76.7 | −0.395 | 0.900 | 177.6 | 0.087 | 0.215 | 45.2 |
| 7 | −0.348 | −0.188 | 462.1 | −0.419 | 0.878 | 181.3 | 0.098 | 0.332 | 151.7 |
| 8 | −0.327 | −0.188 | 77.2 | −0.459 | 0.908 | 183.3 | 0.069 | 0.394 | 151.8 |
| 9 | −0.332 | −0.150 | 474.3 | −0.458 | 0.941 | 49.2 | 0.040 | 0.318 | 160.0 |

As shown in Table A1, rewards gradually increase over iterations across all three reasoning types. In contrast, Epiplexity fluctuates substantially throughout training, without exhibiting the same monotonic trend. This divergence illustrates why monitoring reward alone is insufficient to detect the collapse phenomena discussed in Absolute Zero (Zhao et al., 2025a): a steadily improving reward signal can coexist with highly unstable or degenerate generative complexity.

Moreover, Absolute Zero reports a *reward seesaw* phenomenon in self-play reinforcement learning, which we also reproduce in our experiments. Epiplexity provides complementary insight by indicating whether this seesaw reflects meaningful, learnable structure or merely zero-sum reward redistribution between the proposer and solver. When rewards shift between roles without a corresponding change in Epiplexity, this suggests the system is engaged in reward gaming rather than genuine learning progress.

## E. Data Case

Figures A4, A5, and A6 present sample instances of the three coding tasks in Zhao et al. (2025a).

1. In the abduction task, the model is given the code and the output and is required to predict the input, which demands program analysis and inverse reasoning, inferring which inputs would produce the specified output when executed by the program.

2. In the deduction task, the model receives the code and the input and predicts the output. This is the most common

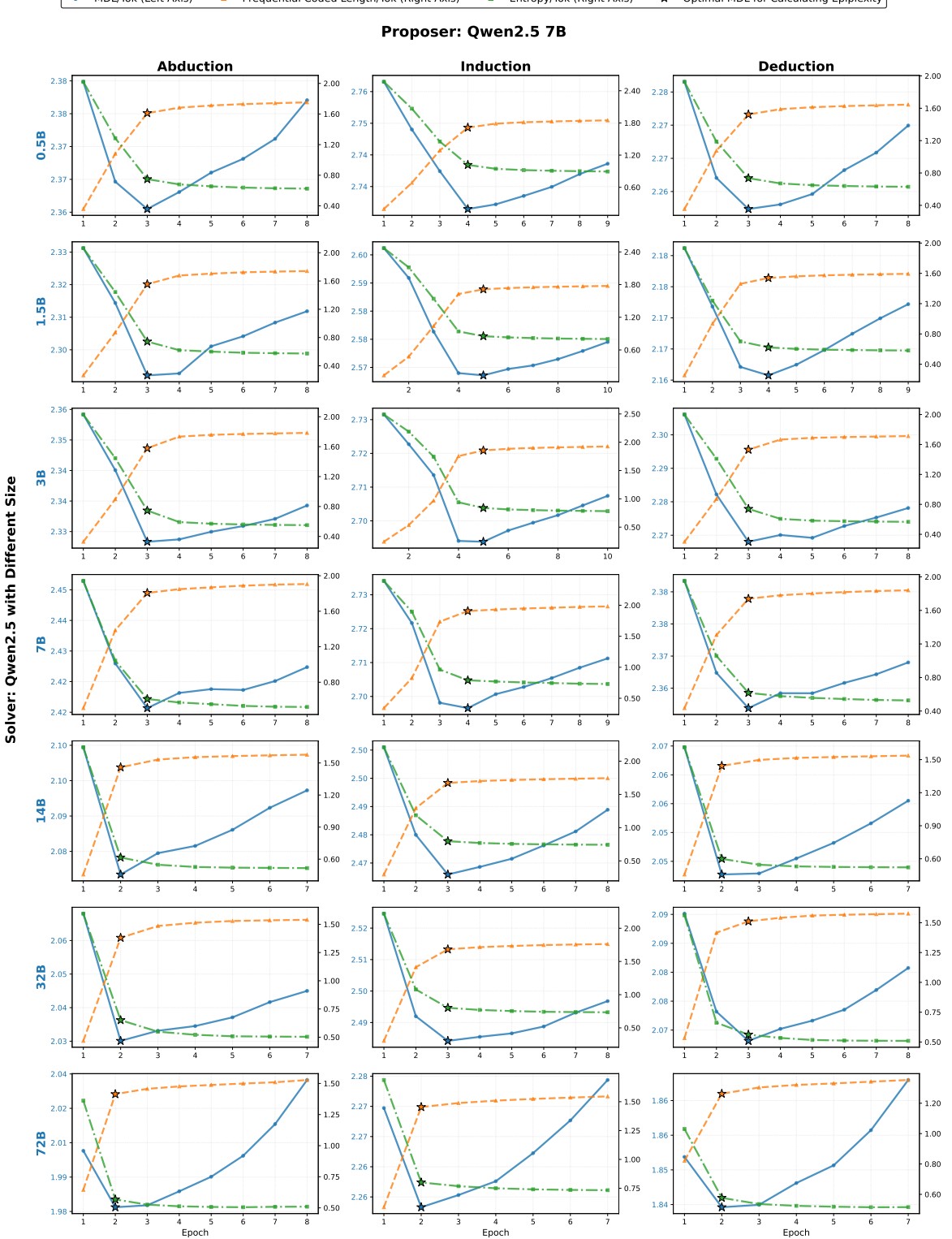

*Figure A1.* Epiplexity calculation visualisation on the synthetic data proposed by Qwen2.5 7B. The ⭐ denotes epiplexity (per-token).

setting, except that the model cannot execute the program in an external environment and must instead analytically derive the output.

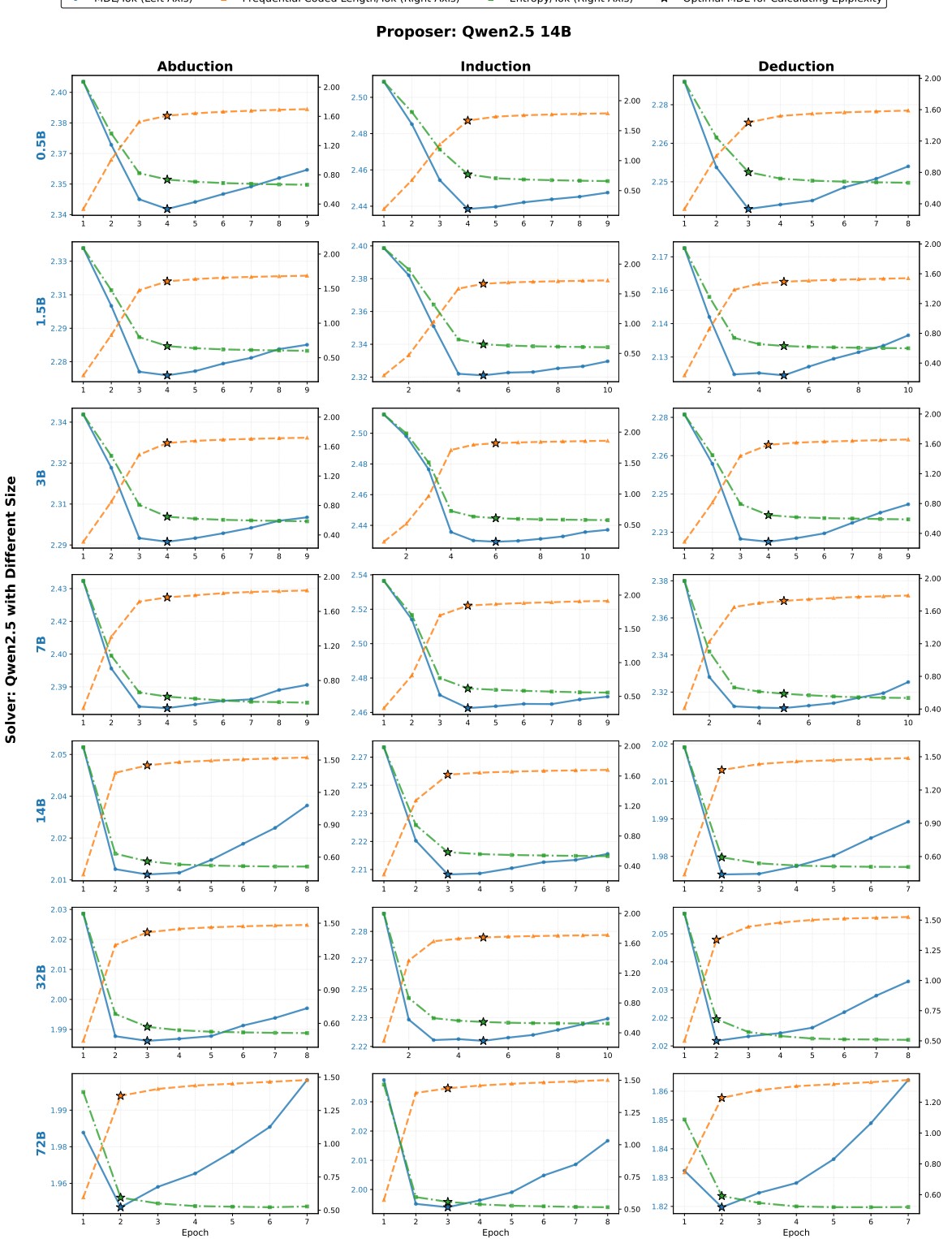

*Figure A2.* Epiplexity calculation visualisation on the synthetic data proposed by Qwen2.5 14B. The ★ denotes epiplexity (per-token).

3. In the induction task, the model is provided only with inputs and outputs and must infer the underlying program. This is the most challenging setting. To ensure the uniqueness of the mapping from input and output to the program, multiple

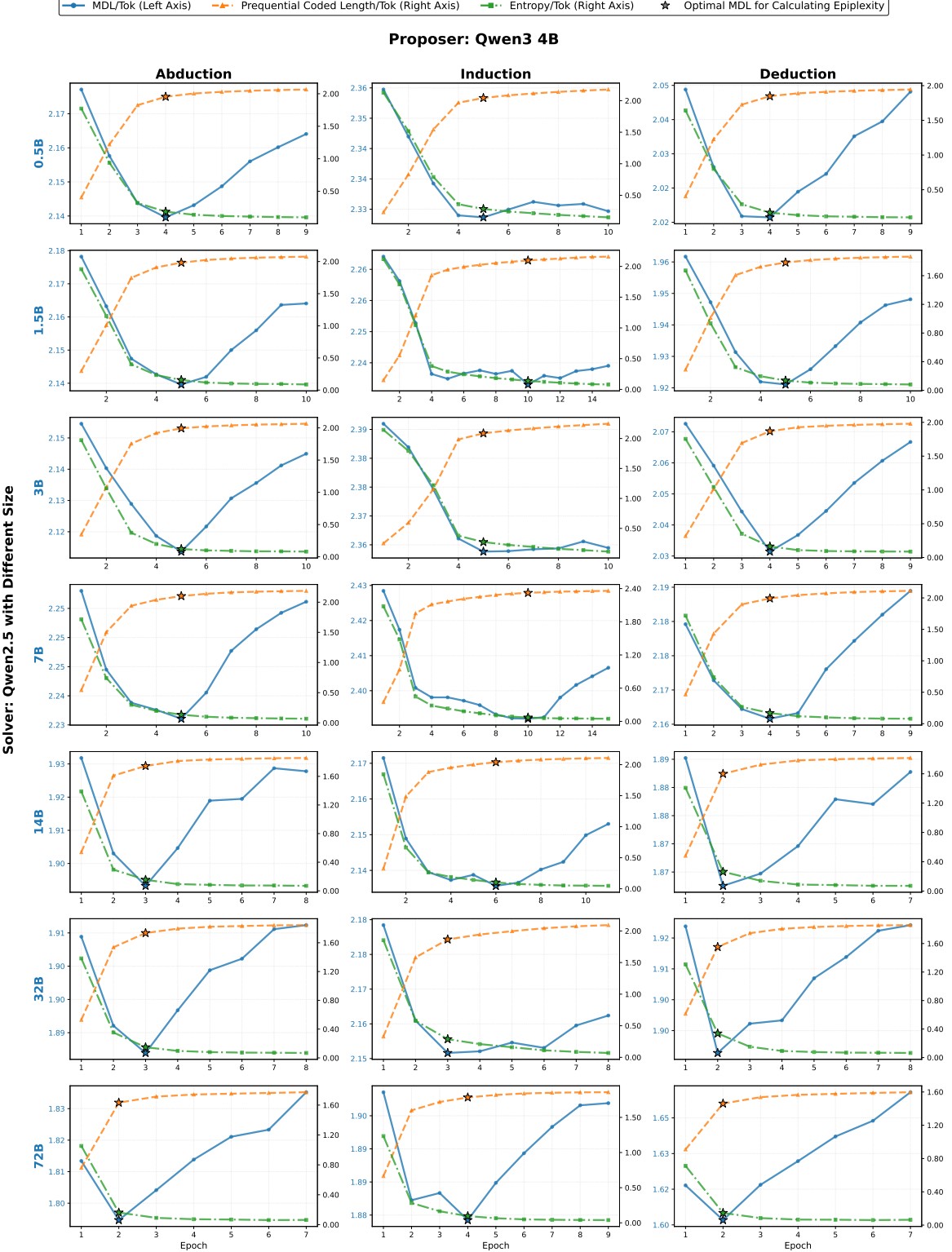

*Figure A3.* Epiplexity calculation visualisation on the synthetic data proposed by Qwen3 4B. The ★ denotes epiplexity (per-token).

input-output pairs are provided. The model must identify the regularities governing the input-output mapping and abstract them into a program.

---

## Data Sample for Abduction

# Task: Provide One Possible Input of a Python Code Snippet Given the Code and Output Given the following Code Snippet and the Output, think step by step then provide one possible input that produced the output. The input needs to be wrapped in ```input``` tags. Remember if an argument is a string, wrap it in quotes. If the function requires multiple arguments, separate them with commas.

# Code Snippet:

```python
def f(nums):
    def transform_list(nums):
        return [num * 2 for num in nums]
    def sum_of_list(nums):
        return sum(nums)
    state = 0
    for _ in range(3):
        nums = transform_list(nums)
        state += sum_of_list(nums)
    return state
```

# Output:

```output
84
```

# Output Format:

```input
arg1, arg2, ...
```

# Example Output:

```input
'John', {'age': 20, 'city': 'New York'}
```

*Figure A4.* Data Sample for Abduction.

---

**Data Sample for Deduction**

# Task: Deduce the Output of a Python Code Snippet Given the Code and Input Given the following Code Snippet and the Input, think step by step then deduce the output that will be produced from plugging the Input into the Code Snippet. Put your output in ```output``` tags. Remember if the output is a string, wrap it in quotes. If the function returns multiple values, remember to use a tuple to wrap them.

# Code Snippet:

```python
def f(nums):
    def sum_of_subarrays(nums):
        total_sum = 0
        for i in range(len(nums)):
            for j in range(i, len(nums)):
                total_sum += sum(nums[i:j + 1])
        return total_sum
    def sum_of_digits(n):
        return sum((int(digit) for digit in str(n)))
    total_sum = sum_of_subarrays(nums)
    return sum_of_digits(total_sum)
```

# Input:

```input
[1, 2, 3, 4]
```

# Example Output:

```output
{'age': 20, 'city': 'New York'}
```

*Figure A5.* Data Sample for Deduction.

---

### Data Sample for Induction

# Task: Deduce the Function that Produced the Outputs from the Inputs Given a set of input/output pairs and a message that describes the function, think through the problem step by step to deduce a general code snippet. This code should produce the hidden outputs from the hidden inputs, matching the original data-generating code that created the input/output pairs. Place your final answer inside python tags! It may be helpful to work through each input/output pair individually to test your function. If your function doesn't work as expected, revise it until it does. The final code snippet will be used to evaluate your response, which is wrapped in ```python``` tags.

# Code Requirements:

- Name the entry function `f` (e.g., `def f(...): ...`), you can have nested definitions inside `f`
- Ensure the function returns a value
- Include at least one input parameter
- Make the function deterministic
- AVOID THE FOLLOWING:
  * Random functions or variables
  * Date/time operations
  * I/O operations (reading files, network requests)
  * Printing or logging
  * Any external state
- Ensure execution completes within 10 seconds on a modern CPU
- All imports and class definitions should be at the very top of the code snippet
- The snippet should end with a return statement from the main function `f()`, anything after will be removed

# Input and Output Pairs:

```input_0
[1, 3, 5]
```
```output_0
8
```
......
```input_4
[22, 24, 26, 28, 30]
```
```output_4
0
```

# Message:

```message
Please analyze the following outputs and try to deduce the function that produces them. Consider the operations performed on the input list and the expected output based on those operations. Think about the intermediate steps and the final result.
```

# Example Output:

```python
def f(a):
    return a
```

Name your entry function `f()`!!!

*Figure A6.* Data Sample for Induction.

