# OpenReview forum: "Position: Self-Play Only Evolves When Self-Synthetic Pipeline Ensures Learnable Information Gain"
_ICML.cc/2026/Position_Paper_Track — ICML 2026 Position Paper Track regular_

### Official Review · Reviewer_myKU · 2026-03-12

**Significance:** 3
**Argument Clarity:** 3
**Rating:** 5
**Confidence:** 4

**Questions:**

1. How sensitive are the epiplexity estimates to choices like tokenization, training hyperparameters selection? Could the observed non-monotonic trends (e.g., stronger solvers eventually decreasing estimated learnable information due to memorization) be artifacts of training regime?

2.  If a system measures learnable information per iteration, then how does the internal environment adapt proposals/verifications (or retrieval policies) to keep the stream in the "Goldilocks zone"?

**Alternative Views Section:**

Yes

**Compliance With Llm Reviewing Policy A Conservative:**

Affirmed.

**Discussion Potential:**

3

**Paper Summary:**

The paper argues that many existing self-play systems reach plateau quickly because they generate more data without increasing learnable information for the next iteration. It frames self-evolution as a triadic loop where one LLM plays PROPOSER, SOLVER, and VERIFIER, and claims progress requires the synthetic-data pipeline to keep producing data in a “Goldilocks zone” of learnable structure, that the structure is complex enough to be non-trivial but structured enough to be learnable. So, the design recommendations are: (1) asymmetric co-evolution, (2) capacity growth, and (3) proactive information seeking.

**Position:**

Yes

**Position In Title:**

Yes

**Related Work:**

3

**Strengths And Weaknesses:**

Strength:
1. The paper points out: self-play loops can keep sampling but fail to increase learnable information for the next iteration, leading to trivialization or instability.
2. Attempts to connect claims to measurable quantities (epiplexity and bounded entropy) and role/capacity effects in coding self-play.
3. Diagnostic experiments aligned with the key ideas. The experiments connect proposer/solver capacities and “synthetic directions” (abduction/deduction/induction) to estimated learnable information.

Weakness:
It will be beneficial to demonstrate how to enforce monotonic learnable information gain in practice. While epiplexity  is presented as a measurement tool, the paper does not fully specify a concrete control strategy that ensures learnable information increases iteration-to-iteration under real training noise and distribution shifts.

**Support:**

3

---

> ### Author Rebuttal · Authors · 2026-03-29
>
> We greatly appreciate your comments! We hope the following clarifications help address your questions.
>
> ---
>
> ### Q1: Perplexity as Measurement Tool
>
> - Yes, we primarily diagnose current self-play RL problems from an information-theoretic perspective, which essentially measures the learning effectiveness of models on self-synthetic data. In the paper, we do not propose a concrete method for optimising toward maximum learnable information, because through our framework analysis, we have identified several clearly missing components. For example, the asymmetric loop has not been closed, the model's computing capacity does not match the data difficulty, and external data is only passively received. Only after these missing components are addressed can we further explore specific algorithms for stabilising information gain.
> - Our primary position in this work is a **call to action** for the self-evolving community to shift attention from focusing solely on RL training to simultaneously attending to learnable information. We will continue to explore concrete methods in future work.
>
> ---
>
> ### Q2: Sensitivity to Hyperparameter Selection
>
> - Epiplexity is a function of both the model and the data, not of the data alone. Therefore, different models (for current LLMs, different tokenisers generally correspond to different models) will yield differences in absolute values, but the relative trends remain consistent.
> - Furthermore, Epiplexity is defined as the MDL-optimal model compression length, so theoretically it corresponds to the hyperparameter-optimal setting.
> - In practice, we adopt Prequential Estimation as shown in the pseudocode in Appendix B. We conducted hyperparameter experiments on Experiment 1 in Section 4 and computed the Coefficient of Variation (CV), as follows:
>
> | Hyperparameter | CV |
> |---|---|
> | LoRA Dropout | 0.9% |
> | Weight Decay | 1.1% |
> | Train/Valid Split Seed | 2.3% |
> | Batch Size | 3.3% |
> | LoRA Rank | 6.0% |
> | Learning Rate | 6.2% |
> | LoRA Target Modules | 14.7% |
>
> For hyperparameters that do not affect model compute capacity, the results are relatively insensitive. For LoRA-related parameters, the impact is larger because they alter the capacity to participate in learning. However, regardless of the experimental setting, the relative trends remain consistent; that is, the trend of Epiplexity with respect to proposer/solver capacity is preserved, and induction data consistently exhibits substantially higher Epiplexity than abduction and deduction.
>
> ---
>
> ### Q3: Concrete Practices for the Identified Gaps
>
> - As noted in Q1, in Sections 3.1, 3.2, and 3.3, we identify three missing components in current self-play RL systems, and each section proposes concrete practices. Specifically,
>     - In Section 3.1, when the asymmetric loop is not closed, the capabilities of the proposer and verifier cannot keep pace with the solver's progress, causing problems to become too easy for the solver and the self-synthetic dataset to fall outside the Goldilocks zone. In the Practice paragraph, we suggest two practical implementations, back-translation and verifier-free RL, to improve the proposer and verifier.
>     - Sections 3.2 and 3.3 also give practice: matching model compute capacity to the current data difficulty, and enabling the model to proactively select external data to introduce fresh information asymmetry, are both aimed at keeping the self-synthetic dataset within the Goldilocks zone at each iteration, thereby ensuring that sufficiently large learnable information can be extracted.

---

> > ### Author Rebuttal · Reviewer_myKU · 2026-04-03
> >
> > Thanks for the clarification! My concerns are resolved and I still keep my acceptance score.

---

### Official Review · Reviewer_fUxz · 2026-03-12

**Significance:** 3
**Argument Clarity:** 3
**Rating:** 4
**Confidence:** 3

**Questions:**

N/A.

**Alternative Views Section:**

Yes

**Compliance With Llm Reviewing Policy A Conservative:**

Affirmed.

**Discussion Potential:**

3

**Paper Summary:**

This position paper argues that many recent “self-evolving” LLM systems are more accurately characterized as self-play systems, and that their main failure mode is not merely reward instability but the lack of sustained growth in learnable information across iterations. The paper frames self-evolution as a triadic loop involving a *proposer*, *solver*, and *verifier*, and argues that genuine long-term improvement requires three ingredients: asymmetric co-evolution, capacity growth, and proactive information seeking. The paper introduces an information-theoretic lens, to support this argument, based on bounded-MDL / epiplexity and presents small-scale self-play coding experiments showing that self-play can yield unstable or collapsing information trajectories rather than monotonic improvement.

**Position:**

Yes

**Position In Title:**

Yes

**Related Work:**

3

**Strengths And Weaknesses:**

**strengths**
- The paper identifies a timely and meaningful problem regarding self-improvement of agentic / foundational models, and the paper is well-written and easy to follow.
- The proposer–solver–verifier perspective is an interesting abstraction that unifies several strands of prior work and makes the system-level dependencies more explicit. In particular, the paper’s view that the proposer and verifier jointly form an ``internal environment" for the solver is conceptually clean and helps explain why improving only one part of the loop is often insufficient.
- The arguments are also backed up by diagnostic, empirical findings which well illustrate how learnable information varies across roles, capacities, and synthetic directions.

**weaknesses**
- One major concern is that the paper's central claim may be stronger than the evidence supporting it, as stagnation in self-evoloving systems may be caused by other factors than failures to increase learnable information, such as reward hacking, verifier drift, collapse of exploration, etc. The paper acknowledges some of these alternatives, but the current evidence does not yet establish learnable-information growth as the dominant explanatory variable.
- Relatedly, the empirical support is still fairly limited. The coding self-play experiments appear diagnostic and illustrative rather than decisive.
- I also found the theoretical layer interesting but somewhat under-validated. The bounded-MDL / epiplexity framing is promising, yet the paper presently uses it more as an interpretive scaffold than as a sharply testable theory. In other words, the formalism helps organize the paper’s perspective, but it is not yet clear how robustly it predicts real system behavior across settings.

**Support:**

3

---

> ### Author Rebuttal · Authors · 2026-03-29
>
> We sincerely thank the reviewer for the positive evaluation of this paper. We hope the following clarifications help address your concerns.
>
> ---
>
> ### Q1: Other Failure Possibilities
> - This is an excellent question! In fact, we believe that from an RL training perspective, all the factors, including reward hacking, verifier drift, and collapse of exploration, are indeed possible causes of self-evolution failure. We argue that these causes manifest in the self-synthetic data pipeline as insufficient learnable information in the self-synthesised data.
> - In the context of self-evolution, information is repeatedly transformed between data and model parameters. Our perspective examines the **output** of this transformation, while the RL training dynamics examines the **transformation method** itself, and these are two complementary viewpoints and do not constitute confounding factors.
> - Our **Call to Action** is to broaden the community's focus beyond RL training dynamics alone, expanding it to examine the problem from the perspective of self-synthetic data.
> - Consider the Absolute Zero example: from a reward-only perspective, it is difficult to monitor whether the self-evolution process remains healthy. We additionally report the proposer and solver rewards along with the corresponding Epiplexity for each of the 9 iterations across the three reasoning tasks in Experiment 2, as shown in the table below. Ind/Abd/Ded = Induction/Abduction/Deduction; P.R/S.R = Proposer/Solver Reward; Epi = Epiplexity (×10³ bits).
>
> | Iter | Ind P.R | Ind S.R | Ind Epi (k) | Abd P.R | Abd S.R | Abd Epi (k) | Ded P.R | Ded S.R | Ded Epi (k) |
> |:----:|:-------:|:-------:|:-----------:|:-------:|:-------:|:-----------:|:-------:|:-------:|:-----------:|
> | 1 | -0.781 | -0.296 | 75.7 | -0.152 | 0.147 | 48.8 | -0.065 | -0.036 | 146.2 |
> | 2 | -0.478 | -0.282 | 76.7 | -0.181 | 0.569 | 49.4 | 0.041 | 0.273 | 156.3 |
> | 3 | -0.428 | -0.273 | 458.9 | -0.297 | 0.733 | 49.2 | 0.045 | 0.240 | 148.9 |
> | 4 | -0.368 | -0.227 | 459.6 | -0.320 | 0.775 | 162.5 | 0.124 | 0.217 | 157.4 |
> | 5 | -0.332 | -0.157 | 76.5 | -0.356 | 0.847 | 50.2 | 0.091 | 0.266 | 45.4 |
> | 6 | -0.360 | -0.197 | 76.7 | -0.395 | 0.900 | 177.6 | 0.087 | 0.215 | 45.2 |
> | 7 | -0.348 | -0.188 | 462.1 | -0.419 | 0.878 | 181.3 | 0.098 | 0.332 | 151.7 |
> | 8 | -0.327 | -0.188 | 77.2 | -0.459 | 0.908 | 183.3 | 0.069 | 0.394 | 151.8 |
> | 9 | -0.332 | -0.150 | 474.3 | -0.458 | 0.941 | 49.2 | 0.040 | 0.318 | 160.0 |
>
> As shown, rewards gradually increase to high values over iterations, while Epiplexity fluctuates substantially. This demonstrates precisely why monitoring reward alone cannot effectively detect the collapse phenomena described in the Absolute Zero paper. Furthermore, the Absolute Zero paper reports a reward seesaw phenomenon in self-play RL, which we also reproduce. Epiplexity tells us whether the seesaw dynamics are producing meaningful learnable structure or merely zero-sum reward redistribution.
>
> ---
>
> ### Q2: Diagnostic Experiments and Under-Validated Experiments
>
> - We acknowledge that, as a position paper, we do not further propose large-scale experiments or methods specifically optimising learnable information. However, by shifting the perspective to re-examine current self-play RL work, we have already identified three system-level gaps that require improvement, and we propose concrete practices for each.
> - Furthermore, we believe that while Epiplexity provides an excellent theoretical perspective, in practical terms, it is not fundamentally different from the best practices already widely adopted in industry for data filtering and synthetic data generation during pre-training. These methods have been well tested, and the community has not yet systematically introduced them into the diagnosis and improvement of self-play RL, which is precisely what we advocate.
> - We really appreciate your suggestion. As a position paper, we aim to open the discussion; in future work, we plan to scale up and conduct concrete, grounded research.

---

> > ### Author Rebuttal · Reviewer_fUxz · 2026-04-04
> >
> > Thank you for the detailed and comprehensive rebuttal, and I have decided to maintain my current rating.

---

### Official Review · Reviewer_Y7XP · 2026-03-13

**Significance:** 1
**Argument Clarity:** 1
**Rating:** 4
**Confidence:** 4

**Questions:**

See weaknesses

**Alternative Views Section:**

Yes

**Compliance With Llm Reviewing Policy A Conservative:**

Affirmed.

**Discussion Potential:**

1

**Final Justification:**

See rebuttal acknowledgement.

**Paper Summary:**

The paper notes that several self-play loops for LLMs either plateau or collapse after a few rounds of self-play. The paper posits that such self-evolving systems, which evolve the proposer, solver and verifier jointly, must cooperate to ensure a monotonic increase in *learnable information.* The paper then goes on to define learnable information in information-theoretic terms and proposes  design principles for such self-play systems:

1. **Asymmetric co-evolution:** between solver and internal environment.
2. **Capacity growth:** including increased model parameters or inference-time compute.
3. **Proactive information seeking:** where the system introduces external context to avoid stagnation.

**Position:**

Yes

**Position In Title:**

Yes

**Related Work:**

1

**Strengths And Weaknesses:**

The paper makes certain claims that are not necessarily correct. For eg., “proposing and verifying are substantially easier than solving“. In several problems in mathematics, it is not easy to verify solutions. For eg, proof writing. Even code writing, it is not easy to ensure that the generated code is correct and passes test cases.

Additionally, while proposing the asymmetric co-evolution design, the authors claim that the verifier scales with the solver and proposer remains at the solver frontier, without proposing any kind of research program or directions to achieve such an aim.

The authors also do not make it clear what precisely are they proposing, besides the use of information-theoretic quantities for measuring information gain, and how the implementable research program is different from what is being done in practice:

1. Many proposed designs such as capacity scaling and external information retrieval are already getting used, as mentioned by the author.

The paper posits curriculum learning and self-play RL as alternatives, without differentiating these methods from their proposed framework for self-play.

Additionally, STaR does not **just** filter correct reasoning traces for finetuning. For incorrectly answered questions, STaR provides either the answer or a hint for another rollout and finetunes on those traces.

Some terms are left undefined:

1. The bounded MDL optimizer contains a term $|P|$, called epiplexity, where $P$ is a a member of a family of LLM observers. The paper does not contain a definition of this term or a motivation as to why such an information theoretic quantity is useful, versus other terms commonly used in bayesian experimental design and data acquisition for training, such as information gain as measured by mutual information or difference in log-likelihoods of model trained on a new set of samples and an older iteration, see [1].
2. PPT is not defined.

[1] Mindermann, Sören, et al. "Prioritized training on points that are learnable, worth learning, and not yet learnt, 2022." URL https://arxiv. org/abs/2206.07137.
[2]

**Support:**

1

---

> ### Author Rebuttal · Authors · 2026-03-29
>
> We greatly appreciate your suggestions and are delighted that you are willing to discuss this paper from the perspectives of information-theoretic metrics and data selection. We hope the following responses address your concerns.
>
> ### Q1: Proposing and Verifying Are Substantially Easier Than Solving
> - We fully agree with this observation. Of course, not all tasks in all domains (even mathematics and coding) exhibit the property that proposing and verifying are easier than solving. **In fact, we have clearly articulated this point in the paper**, including the math proof example. **Please refer to the Practice paragraph in Section 3.1 and Figure 2**. We explicitly identify that the asymmetry gap can be positive (e.g., general math problems), near-zero (e.g., grammar correction), or negative (e.g., healthcare, policymaking). Even within a single domain (math), the gap can be positive (e.g., Sudoku), near-zero (e.g., large-number multiplication), or **negative (e.g., mathematical proof)**.
> - Our position in the paper is that one should not focus solely on the easy-to-verify side, which is the limitation of the vast majority of current self-play RL work, but also orchestrate tasks according to their asymmetry gap and close the asymmetric loop, thereby enabling healthy co-development on both sides of the asymmetry.
> ---
> ### Q2: Practice on Proposer and Verifier Improves with the Solver
> - We provide a **Practice** paragraph in Section 3.1 (as well as in Sections 3.2 and 3.3).
> - Please refer to lines 220–240, where we explicitly suggest that proposers can improve alongside solvers via *back-translation*, and verifiers can improve alongside solvers via *verifier-free RL*.
> ---
> ### Q3: Missing Position and No distinct implementation plan
> - **Our position is presented in the Introduction, Sections 3.1, 3.2, 3.3, and the Call to Action in Section 8.** We call on the community to move beyond viewing self-evolution solely from the perspective of RL, and instead analyse it through the lens of the learnable information contained in self-synthesised data. Within this analytical framework, we identify several system-level deficiencies in current mainstream self-play RL that require attention, as discussed in Section 3.
> - All practices we propose had not been explored in the self-evolving LLM at submission time.
> - These practices are grounded in existing techniques to be implementable and are derived directly from the gaps identified through our information-theoretic analysis.
> ---
> ### Q4: Difference Between Self-Play RL and Curriculum Learning
> - **The paper does not propose any self-play framework**. As described in our abstract, instead, it reorganises existing work into three roles and introduces an information-theoretic perspective to explain failure modes.
> - Regarding the difference compared with curriculum learning, we explicitly note in the **Alternative Views** section. An intuitive distinction is that for curriculum learning, we arrange data according to the model’s capabilities, whereas in our framework, the model and data co-evolve to extract the maximal amount of learnable information.
> ---
> ### Q5: Description of STaR
> - We acknowledge that our description was not sufficiently precise. A more accurate characterisation is that STaR filters correct reasoning traces and recycles rationalised ones for fine-tuning.
> ---
> ### Q6: Definition of P
> - The family of observer instances consists of models that reach a fixed compute budget through different parameter counts, active parameters, or inference-time computation, as described in Section 3.2.
> ---
> ### Q7: Epiplexity vs Other Metrics
> - We do not claim that epiplexity is more useful than other indicators (including the rho-loss you mentioned), because our position is to examine self-play RL from the perspectives of information theory and computational complexity, identifying which system-level components are still missing for achieving sustainable self-evolution. **We have never claimed to propose a superior information-theoretic metric (Epiplexity itself was not proposed by us)**. This is a position paper instead of a paper proposing a new method or a new metric.
> - Metrics such as rho-loss are also valuable. Epiplexity offers an additional perspective: learnable information is a function of both the observer model's computation and the data, rather than of the data alone. This directly motivates our claim in Section 3.2 that the compute budget needs to grow dynamically across iterations.
> ---
> ### Q8: PPT Is Not Defined
> - PPT refers to Probabilistic Polynomial-Time. We apologise for the omission.

---

> > ### Author Rebuttal · Reviewer_Y7XP · 2026-04-04
> >
> > I thank the authors for the clarifications they offer, but I still do not see a concrete implementable program beyond that what has already been proposed and being acted upon.

---

### Official Review · Reviewer_hbzR · 2026-03-15

**Significance:** 3
**Argument Clarity:** 3
**Rating:** 4
**Confidence:** 3

**Questions:**

1. Why the learnable information increases in the order of “from Qwen2.5 7B to Qwen2.5 14B to Qwen3 4B”? Would there be any detailed explanations that could directly correspond to the theoretical framework?

2. relevance between the reward and  in the demo task. How do the reward curves look like in the experiments? How would the epiplexity-reward relevance change the position?

**Alternative Views Section:**

Yes

**Compliance With Llm Reviewing Policy A Conservative:**

Affirmed.

**Discussion Potential:**

3

**Final Justification:**

I believe the paper is currently in the condition of acceptable and hence I will maintain my current score of 4.

**Paper Summary:**

This paper tries to identify the root of the failures of the existing LLM iterative improvement pipeline with synthetic data, i.e., the missing of learnable information monitoring. To better evaluate learning bound, they propose to use the minimum description length (MDL)  as the theoretical ground to decompose learnable structure information and . To further make the measurement practical, they introduces epiplexity as a compute-conditional metric for the reusable information that needs to learn.

By re-formalizing the self-evolved into self-play form, the authors a posit the pipeline 3-component system (prover, solver and verifier), allowing to cover the use cases more comprehensively: RLVR, open-ended RL, and instruction tuning, etc. By combining the MDL measurement with the new system view, the authors design two experiments and derive three principles for the design of future self-play algorithms: asymmetric co-evolution, capacity growth and proactive information seeking.

**Position:**

Yes

**Position In Title:**

Yes

**Related Work:**

4

**Strengths And Weaknesses:**

Strengths

1. The position paper provide solid theoretical grounding, using MDL equipped with epiplexity as an quantified measurement of the compressible and unlearnable information contained in the synthesized data.

2. The proposed framework and the comparison in alternative view part provides clear insights and guidance for practical improvement (especially the curriculum learning part).

Weaknesses:

1. misalignment of the metric: the experiment part solely shows the expiplexity values under different settings but missing the reward. As the authors have argued that the reward is not sufficient/in-appropriate for judgement, they do not provide concrete proof of this argument. This weakens the conclusions derived from the experiment.

2. lack of improvement experiment: while posing the guess of failure reasons, this work does not given they . Although the framework seems to help the community to avoid monotonic increasing of model size, the paper does not provide substantial and specific evidence of the usefulness of the practice guide (even in demo-level).

**Support:**

3

---

> ### Author Rebuttal · Authors · 2026-03-29
>
> We sincerely thank the reviewer for the positive evaluation of this paper. We hope the following clarifications help address the concerns.
>
> ---
>
> ### Q1: Misalignment of the Metric
>
> - To clarify the relationship between reward and Epiplexity, we report the proposer and solver rewards together with Epiplexity across the nine iterations of Experiment 2. Ind/Abd/Ded denote Induction, Abduction, and Deduction; P.R/S.R denote Proposer/Solver Reward; Epi denotes Epiplexity (×10³ bits).
>
> | Iter | Ind P.R | Ind S.R | Ind Epi (k) | Abd P.R | Abd S.R | Abd Epi (k) | Ded P.R | Ded S.R | Ded Epi (k) |
> |:----:|:-------:|:-------:|:-----------:|:-------:|:-------:|:-----------:|:-------:|:-------:|:-----------:|
> | 1 | -0.781 | -0.296 | 75.7 | -0.152 | 0.147 | 48.8 | -0.065 | -0.036 | 146.2 |
> | 2 | -0.478 | -0.282 | 76.7 | -0.181 | 0.569 | 49.4 | 0.041 | 0.273 | 156.3 |
> | 3 | -0.428 | -0.273 | 458.9 | -0.297 | 0.733 | 49.2 | 0.045 | 0.240 | 148.9 |
> | 4 | -0.368 | -0.227 | 459.6 | -0.320 | 0.775 | 162.5 | 0.124 | 0.217 | 157.4 |
> | 5 | -0.332 | -0.157 | 76.5 | -0.356 | 0.847 | 50.2 | 0.091 | 0.266 | 45.4 |
> | 6 | -0.360 | -0.197 | 76.7 | -0.395 | 0.900 | 177.6 | 0.087 | 0.215 | 45.2 |
> | 7 | -0.348 | -0.188 | 462.1 | -0.419 | 0.878 | 181.3 | 0.098 | 0.332 | 151.7 |
> | 8 | -0.327 | -0.188 | 77.2 | -0.459 | 0.908 | 183.3 | 0.069 | 0.394 | 151.8 |
> | 9 | -0.332 | -0.150 | 474.3 | -0.458 | 0.941 | 49.2 | 0.040 | 0.318 | 160.0 |
>
> - As shown, rewards gradually increase over iterations. In contrast, Epiplexity fluctuates substantially. This illustrates why monitoring reward alone is insufficient to detect the collapse phenomena discussed in Absolute Zero.
> - Moreover, Absolute Zero reports a reward seesaw phenomenon in self-play RL, which we also reproduce. Epiplexity provides complementary insight by indicating whether this seesaw reflects meaningful learnable structure or merely zero-sum reward redistribution.
> ---
>
> ### Q2: Lack of Improvement Experiment
>
> - Our goal in this work is primarily to encourage a shift in perspective, from focusing solely on RL design to examining the self-synthetic data pipeline. Under this lens, we identify several gaps that remain largely unaddressed: the asymmetric co-evolution loop is incomplete, model capacity does not grow dynamically, and external information is not actively explored.
> - For each gap, the paper provides both an information-theoretic interpretation and a dedicated Practice section outlining concrete directions. Once these gaps are addressed, we plan to further explore algorithms that ensure sufficient learnable information at each iteration. As a position paper, we therefore focus on diagnostic analysis and aim to stimulate broader discussion.
>
> ---
>
> ### Q3: Why Learnable Information Increases as Proposer Ability Increases
> - Proposers of different capabilities draw from different information sources (i.e., the proposer model's own knowledge) to generate synthetic tasks. A stronger proposer possesses richer internal knowledge;
> - Moreover, even given the same information source, a more capable model can perform more effective transformations, injecting more learnable information into the synthetic data. Consequently, the solver model, acting as the observer, can extract more learnable information from these tasks. As shown in Section 3.1, for the same information source, different transformations yield different amounts of learnable information.

---

> > ### Author Rebuttal · Reviewer_hbzR · 2026-04-04
> >
> > I thank the authors for their detailed response and for providing the missing reward data. The comparison table between Reward and Epiplexity in Experiment 2 is particularly illuminating; it clearly demonstrates that while task-level rewards may show monotonic improvement, the underlying learnable information can fluctuate or collapse . This directly addresses my concern regarding the "misalignment of the metric" and provides empirical support for the authors' position that reward optimization is a necessary but insufficient criterion for genuine self-evolution.
> >
> > Regarding the lack of an improvement experiment, I appreciate the authors' clarification that this work is primarily a position paper intended to diagnose systemic gaps and provide a theoretical roadmap . The "Practice" sections indeed offer concrete directions for addressing these gaps, such as asymmetric scaling and adaptive reasoning length . However, as the authors acknowledge, realizing a truly robust self-evolving system remains a major challenge. While the diagnostic analysis is now much stronger, the practical utility of the proposed principles in fixing the observed collapse in a closed-loop setting remains to be fully demonstrated in future algorithmic work.
> >
> > Overall, the rebuttal has strengthened the paper's core thesis, though the transition from diagnosis to a proven solution remains a significant open research problem.

---

### Decision · Program_Chairs · 2026-04-30

**Decision:**

Accept (regular)

**Comment:**

Reviewer hbzR noted that their primary concerns had been addressed and they agree that the contribution meets the scope of a position paper, but they would have liked to see a practical implementation showing the value of the proposed system.

Reviewer Y7XP was originally negative about the paper due to concerns that there was not a practical implementation for the proposed improvements to the self-evolution framework. However, the rebuttal seems to have satisfied most of their objections and they acknowledged that the guidelines for position papers do not necessitate practical implementation. Their score increase from 2 to 4 reflects alignment with position paper track expectations rather than a fundamental change in their assessment of the work.

Reviewer fUxz had similar concerns to the above reviewers about practical implementation and evidence that the epiplexity-based approach could add value over the alternate views, though similarly to other reviewers, the rebuttal was sufficient to keep them in borderline accept range.

Reviewer myKU was the most positive about the paper and was satisfied by the rebuttal. Their concerns overlapped with those of the other reviewers including pointing out the lack of a concrete implementation.

The authors' rebuttal provided useful additional empirical data, including a table comparing reward trajectories against epiplexity across 9 self-play iterations, which helped demonstrate that reward monitoring alone cannot detect the collapse phenomena they describe. This strengthened the diagnostic value of their framework.

In light of reviewers asking about practical implementations, I think it is worth the authors considering providing a toy (synthetic/theoretical?) experiment showing usefulness of applying the principles for the final paper (though this should not gate acceptance). The authors noted that there are gaps in current methodology that need to be solved before a true implementation can be introduced in order to stabilize information gain. However, it could be useful if the authors were able to provide some evidence showing that self-evolution can be more effective if certain mitigations are addressed. It is good though that they pointed to techniques such as back-translation and verifier-free RL that can help close the gap.
Side note: it could be interesting to connect to literature on GAN and other adversarial systems where care is needed to ensure the generator and discriminator (or actor and critic, etc) can co-evolve in a way that keeps learning from collapsing. The mode collapse phenomenon in GANs is a natural analogy to the information collapse described in this work.

Also, I think it would be good for the authors to include some review of openendedness literature stemming from works such as the POET algorithm, AlphaEvolve, etc., that are adjacent to this line. While the paper cites GenEnv on environment co-evolution, foundational work on novelty search and quality-diversity methods would strengthen the discussion of how systems can avoid convergence to trivial solutions.

One additional note: the epiplexity metric central to their analysis comes from very recent work (Finzi et al., 2026) and isn't yet widely validated, which partially explains some reviewer hesitance about the theoretical grounding. This is acceptable for a position paper introducing a new lens, but the authors should acknowledge this limitation.